# Hyperhomocysteinemia Induced by Methionine Excess is Effectively Suppressed by Betaine in Geese

**DOI:** 10.3390/ani10091642

**Published:** 2020-09-12

**Authors:** Zhi Yang, Yu Yang, Jinjin Yang, Xiaoli Wan, Haiming Yang, Zhiyue Wang

**Affiliations:** 1Joint International Research Laboratory of Agriculture and Agri-Product Safety of the Ministry of Education of China, Yangzhou University, Yangzhou 225009, Jiangsu Province, China; zhiyang@yzu.edu.cn; 2College of Animal Science and Technology, Yangzhou University, Yangzhou 225009, Jiangsu Province, China; yzuyangyu@163.com (Y.Y.); jjyang777@163.com (J.Y.); wanxl1021@126.com (X.W.); hmyang@yzu.edu.cn (H.Y.)

**Keywords:** Hyperhomocysteinemia, methionine excess, betaine, geese

## Abstract

**Simple Summary:**

Methionine is a proteogenic sulfur amino acid with a vital role in intermediary metabolism. However, excess Methionine (Met) intake is toxic, leading to hyperhomocysteinemia. Betaine supplementation effectively ameliorates biochemical abnormalities. However, a lack of genetic information hinders the understanding of the mechanisms underlying methionine excess-mediated effects and whether Bet can effectively suppress these effects in geese. This study was performed to evaluate the effects of excess methionine on growth performance, serum homocysteine levels, apoptotic rates, and Bax and Bcl-2 protein levels in geese and to study the role of betaine in relieving excess Met-induced hyperhomocysteinemia. It was found that excess methionine reduces body weight induced by myocardial apoptosis, and betaine can be used to effectively lower plasma homocysteine levels.

**Abstract:**

The objective of our study was to investigate the effects of excess Methionine (Met) on the growth performance, serum homocysteine levels, apoptotic rates, and Bax and Bcl-2 protein levels in geese and to study the role of Bet (betaine) in relieving excess Met-induced hyperhomocysteinemia (HHcy). In this study, 150 healthy male 14-day-old Yangzhou geese of similar body weight were randomly distributed into three groups with five replicates per treatment and 10 geese per replicate: the control group (fed a control diet), the Met toxicity group (fed the control diet +1% Met), and the Bet detoxification group (fed the control diet +1% Met +0.2% Bet). At 28, 49, and 70 d of age, the geese in the Met toxicity group had significantly lower body weights than those in the control group (*p* < 0.05). The serum homocysteine levels in geese at 70 d of age in the detoxification group were significantly lower than those in the Met toxicity group (*p* < 0.05). Compared with the control, Met significantly increased cardiomyocyte apoptosis rates, while Bet reduced them. In conclusion, our results suggest that excess methionine reduces body weight induced by myocardial apoptosis, and Bet can be used to effectively lower plasma homocysteine levels.

## 1. Introduction

Methionine (Met) is a sulfur amino acid (SAA) with a vital role in intermediary metabolism [1,2]. Observations made in some prior studies have indicated that optimal supplementation with Met can improve growth performance and body protein synthesis in growing birds [3,4,5]. However, among the constituent amino acids of proteins, Met is toxic if consumed in excess [6]. The supplementation of voluntarily consumed food with Met levels that are four to six times the estimated requirements can suppress food intake and growth [6,7]. The tolerable upper limit of Met for growing ducks were less than 1.38%, considering the growth performance [6]. Excessive Met intake can also cause hepatic oxidative/nitrosative damage, hepatic encephalopathy, erythrocyte morphological alterations, and resultant splenic hemosiderosis in rats [8].

Homocysteine (Hcy) is a nonproteogenic SAA generated by the intrahepatic transmethylation of dietary Met [9]. Hcy functions as a product and a substrate of Met. Hcy accumulation as a result of impaired metabolism leads to an acquired metabolic anomaly known as hyperhomocysteinemia (HHcy), which was first identified by McCully [10]. HHcy occurs when Met levels increase in the blood due to various conditions. Costa et al. [11] demonstrated that dietary supplementation of Met and/or Met sulfoxide alters lipid peroxidation and carbonyl content in young rats.

The feature of atherosclerotic lesions is apoptotic cell death in both animals and humans [12,13]. It has been reported that apoptosis leads to a greater lesion rupture risk by reducing the number of viable smooth muscle cells of mice [14]. However, the cellular pathways responsible for this effect and their correlation with atherothrombotic disease are not fully understood. Recent studies have demonstrated that Hcy causes endothelial cell dysfunction and induces apoptotic cell death among cell types relevant to atherothrombotic disease [15], including endothelial cells and smooth muscle cells. However, the oral administration of betaine (Bet) [16] and B vitamins [17] can alleviate HHcy and attenuate atherogenesis in animal models of rats [17]. Moreover, numerous studies have shown that the process of apoptosis is mediated by proteins of the Bcl-2 family, which comprise antiapoptotic components (e.g., Bcl-2 and Bcl-xL) and proapoptotic components (e.g., Bax, Bak, and Bad). The ratio between B-cell lymphoma 2 (Bcl-2) and Bcl-2 associated protein X (Bax) determines cell survival or death following an apoptotic signal [18].

Excess Met causes marked growth depression in birds. In general, considerable evidence has revealed that excess Met is toxic to many vertebrates, including humans [10], mice [19], and poultry [6,20]. It appears that excess Met could lead to HHcy and reduced body weight by myocardial apoptosis. The biochemical effects of excess Met are complex and are not completely understood, and their mechanisms remain to be elucidated. In particular, information on the metabolic events linking Met excess to pathology in geese is lacking. Hence, many attempts have been made to alleviate HHcy through lower plasma Hcy concentration by folate or Bet, since Bet would promote the metabolism of Hcy and reduce the intravital Hcy level. Considering that the pathophysiology of HHcy is as yet not clear and that high tissue levels of Met may be harmful, the objective of our study was to investigate the effects of excess Met on growth performance, serum Hcy levels, apoptotic rates, and Bax and Bcl-2 protein levels in geese and to study the role of Bet in relieving excess Met-induced HHcy.

## 2. Materials and Methods

### 2.1. Ethics Statement

The animal study was approved by the Institutional Animal Care and Use Committee (IACUC) of the Yangzhou University Animal Experiments Ethics Committee under permit number SYXK(Su) IACUC 2019-0029. All geese experimental procedures were performed according to the Regulations for the Administration of Affairs Concerning Experimental Animals approved by the State Council of the People’s Republic of China.

### 2.2. Experimental Design, Diets, and Management

This study was conducted with 150 healthy male Yangzhou geese at 14 d of age from a commercial hatchery (Yangzhou Goose Co. Ltd., Yangzhou, China). All of the geese had similar body weights (BWs) (473 ± 0.36 g) and were randomized into three groups that included five replicates per treatment and ten geese per replicate. A basal corn/soybean meal diet with two stages (14–28 d and 29–70 d) was formulated to meet the nutritional needs of the geese National Research Council (NRC), 1994 (Table 1). The control group received only the basal diet from 14 d to 70 d of age. The Met toxicity group received the basal diet supplemented with 1% Met. The Bet detoxification group received the diet given to the Met toxicity group supplemented with 0.2% Bet. The supplementation ratios of Met and Bet were cited by Xie et al. [6] and Setoue et al. [16]. The tolerable upper limit of dietary Met for growing ducks was less than 1.38% on the basis of the growth performance. The Met used in the feed was *DL*-Met. The geese were fed in separate plastic-floored pens with 2 cm^2^ square holes that were laid 70 cm above the ground. All manure was cleaned at the end of the trial. All geese were fed and watered *ad libitum* for 56 d. Water was provided in a half-open plastic cylindrical water tank, and the feed was provided in feeders on one side of each pen. The geese were reared indoors under similar environmental conditions (temperature: 26.0 ± 3.0 °C; relative humidity (RH): 65.5 ± 5.0%; lighting period: 16 h; space allocation: 0.5 m^2^/gander).

### 2.3. Sample Collection and Analyses

Feed intake (FI) by pen was measured on a daily basis, and BW was recorded at 14, 28, 49, and 70 d of age. The average daily feed intake (ADFI), average daily gain (ADG), and feed-to-gain ratio (F/G) were calculated from 14 to 28 d and from 29 to 70 d of age, and mortality was recorded as it occurred.

Calculations:

ADFI = feed consumption/feeding days in the whole period

ADG = total weight gain/feeding days in the whole period

F/G = ADFI/ADG

When the geese reached 70 d of age, 2 geese from each treatment replicate (5 replicates per treatment; n = 30 geese) were randomly selected for blood collection from their wing veins.

### 2.4. Clinical Blood Parameters

Blood drawn from wing veins was cooled in ice water and centrifuged for 10 min at 4500 rpm to obtain plasma for the measurement of biochemical indices. The plasma was stored at −20 °C until analysis. The plasma Hcy concentrations in peripheral blood were determined according to the previously described methods of Feussner et al. [21]. The thiol compound was liberated from plasma proteins by reduction with tri-n-butyphosphine and derivatized with a thiol-specific fluorogenic marker, 7-fluoro-benzo-2-oxa-1,3-diazole-4-sulpho-nate(Shanghai Yishi Chemical Co., Ltd, Shanghai, China). The derivative was separated isocratically within 7 min by reversed-phase HPLC using a Superspher 100 RP-18 column(Guangzhou Xinghe Biotechnology Co., Ltd, Guangzhou, China) as stationary phase.

### 2.5. Apoptosis Assay

An annexin V apoptosis detection kit (BD Biosciences, Cat. No. 556547, Lake Franklin,New Jersey, NJ USA) was utilized to measure apoptosis in heart tissue following the manufacturer’s instructions. After treatments, cardiomyocytes were washed twice with cold phosphate-buffered solution (PBS), trypsinized, and then resuspended in binding buffer at a concentration of 1 × 10^6^ cells/mL. Then, aliquots of 100 μL of cell suspension (1 × 10^5^ cells) were incubated with fluorescein isothiocyanate (FITC)-annexin V(Shanghai Yishi Chemical Co., Ltd, Shanghai, China) and propidium iodide for 15 min at room temperature in the dark. The apoptotic rate was analyzed using flow cytometry(Thermo Fisher Technology (China) Co., Ltd, Shanghai, China) within 1 h.

### 2.6. Western Blot Analysis

Total protein lysates were collected for standard immunoblot analysis. The protein concentrations were determined by bicinchoninic acid (BCA) protein assay(Thermo Fisher Technology (China) Co., Ltd, Shanghai, China). Aliquots of protein lysates (30 mg/lane) were loaded into gels for sodium dodecyl sulfide-polyacrylamide gel electrophoresis (SDS-PAGE), and the separated proteins were transferred to a polyvinylidene fluoride (PVDF) membrane(Thermo Fisher Technology (China) Co., Ltd, Shanghai, China). The membrane was blocked and incubated with Bcl-2 and Bax antibodies overnight at 4 °C. The Bcl-2 and Bax antibodies were purchased from AmyJet Scientific (Wuhai, China) (Bcl-2, Cat. No. 3033-100; Bax, Cat. No. 3032-100). The blots were washed with TBST (Sigma-Aldrich, Shanghai, China) and incubated with corresponding horseradish peroxidase-conjugated secondary antibodies (Jackson Laboratory, Bar Harbor, Maine, ME, USA). Finally, the blots were visualized with enhanced chemiluminescence and quantified by densitometry.

### 2.7. Statistical Analysis

The data are expressed as the means ± standard deviations (SDs) and were subjected to Kolmogorov-Smirnov (KS) testing to confirm normality. Different groups were compared using one-way analysis of variance (ANOVA) followed by the Student-Newman-Keuls post hoc test with SPSS 17.0 (SPSS, Shanghai, China). *p* < 0.05 was considered to indicate statistical significance.

## 3. Results

### 3.1. Growth Performance

The data for mortality at 70 d were transformed before analysis. Mortality was greater in the Met toxicity group than in the control group and the Bet detoxification group (*p* > 0.05), and there was no treatment effect between groups. The effects of Bet on the growth performance of geese with HHcy are shown in Table 2. The BWs of the geese at 28, 49, and 70 d of age were significantly lower in the Met toxicity group than in the control group (*p* < 0.05). There was no significant difference in BW at 70 d between the Met toxicity group and the Bet detoxification group (*p* > 0.05). The ADFI and ADG for geese at 14–28 d of age were significantly higher in the Met toxicity group and the Bet detoxification group than in the control group (*p* < 0.05).

### 3.2. Clinical Blood Parameters

The effects of Bet on serum biochemical indices of HHcy in geese are shown in Table 3. The Hcy levels in geese at 28 d, 49 d, and 70 d of age were significantly higher in the Met toxicity group than in the control group (*p* < 0.05). The Hcy levels in geese at 70 d of age in the detoxification group were significantly lower than those in the Met toxicity group (*p* < 0.05) but were not significantly different from those in the control group (*p* > 0.05). The serum Hcy level was a sensitive index for the identification of geese with HHcy. The Hcy values for the geese in the control group were all between 16 and 21 μmol/L, while the Hcy values for the geese in the Met toxicity group were all higher than 30 μmol/L.

### 3.3. Cardiomyocyte Apoptosis

The effects of Bet on cardiomyocyte apoptosis in geese with HHcy are shown in Figure 1. Compared with the control diet, the Met toxicity diet significantly increased cardiomyocyte apoptosis rates (*p* < 0.05), but the Bet detoxification diet reduced the apoptosis rates.

### 3.4. Expression of Apoptosis-Related Proteins

The effects of Bet on the expression of apoptosis-related proteins in geese with HHcy are shown in Figure 2 and Table 4. The protein expression levels of Bcl-2 were significantly lower in the Met toxicity and Bet detoxification groups than in the control group (*p* < 0.05) (Table 4). There were no significant differences between the control group and the other groups regarding the expression of other apoptosis-related proteins.

## 4. Discussion

Excess Met causes marked growth depression in birds, and very high levels of Met can lead to HHcy. However, several studies have suggested that dietary Bet might alleviate HHcy. In the present study, we investigated the effect of Bet on Met loading-induced elevations in plasma Hcy levels. We first established a model of HHcy in geese via Met loading. In previous Met loading studies, Met has been administered orally to humans [22,23] and rats [13].

In this study, the BWs of geese at 28, 49, and 70 d of age were significantly lower in the Met toxicity group than in the control group, and Bet did not improve growth performance in geese at 28, 49, and 70 d of age. In previous studies, excess levels of Met or a Met hydroxy analog have been found to be toxic and to cause growth depression in ducks [24] and broiler chickens [25]. Xue et al. [24] have also reported that excess *DL*-Met and excess L-Met are toxic to starter Pekin ducks and that both Met sources exert equivalent growth-depressing effects. Our results are in agreement with the findings of many previous studies on the toxicity of Met and its hydroxy analog in broilers [25]. At present, little information about Met toxicity in geese has been reported. Our findings also suggest that poultry, especially geese, are intolerant of excess Met from 14 to 70 d of age given that growth performance was continuously depressed during this timeframe. Previous studies have shown that betaine is effective in suppressing methionine-induced HHcy in humans [22] and rats [26]. It is usually assumed that dietary Bet affects Hcy metabolism by stimulating Hcy remethylation. In contrast, we did not detect any strong effect of Bet in Met-induced HHcy model in growth performance in our study. We could still see the Bet could alleviate HHcy-mediated BW reductions in geese at this later growth stage, as there was no significant difference in BW between the Bet detoxification group and the Met toxicity group at the age of 70 d. Possible explanations of the discrepancy are the difference in the time of Met loading: one-shot methionine injection vs. successive methionine feeding. Our test results are consistent with those of Setoue et al. [16].

Hcy is a nonessential, thiol-containing and potentially cytotoxic 4-carbon α-amino acid formed through the demethylation of Met during Met metabolism [26]. Traditional explanations of the mechanism of Hcy neurotoxicity have suggested that disturbances in methylation and remethylation processes play key roles. Increasing evidence supports the hypothesis that elevated total Hcy is an independent risk factor for coronary vascular and neurodegenerative disease [27]. High levels of Hcy are related to cerebrovascular disease, monoamine neurotransmitter deficiencies, and mood depression. Specifically, it has been hypothesized that the cerebrovascular disease and neurotransmitter deficiency caused by high Hcy levels lead to the depression of mood [28]. Taken together, the findings indicate that the serum Hcy level is a sensitive index for the identification of geese with HHcy (HHcy: higher than 30 μmol/L; normal: 16–21 μmol/L) and that 1.35% *DL*-Met supplementation can lead to HHcy in geese. Hyperhomocysteinemia (HHcy) was defined as mild HHcy (Hcy: 15–30 μmol/L), moderate HHcy (Hcy: 30–100 μmol/L), and severe HHcy (Hcy > 100 μmol/L) by measuring the level of Hcy in human blood [29,30]. The results in geese were approximately moderate HHcy in human.

Several studies have suggested that dietary Bet might be a determinant of plasma Hcy concentrations [31,32,33]. Setoue et al. [16] found that dietary supplementation with Bet at a level of 0.34% significantly reduced plasma Hcy levels in rats with HHcy. In the present study, dietary supplementation with 0.2% Bet was markedly effective in decreasing plasma Hcy concentrations in geese with Met loading-induced HHcy at 70 d of age. This finding indicates that Bet should be fed for at least one month to relieve HHcy.

The Hcy-lowering effect of Bet can most likely be ascribed to an increase in Bet-dependent remethylation owing to increased Bet availability and enhanced activity of the enzyme Bet methyltransferase in the liver and kidneys. This possibility is supported by the results of studies by Barak et al. [34] and Saarinen et al. [35], who found that Bet feeding elevates hepatic Bet pools in rats and chickens. Finkelstein et al. [36] have reported that an excessive intake of Met also enhances Bet methyltransferase activity, and Met and Hcy infusion reduce hepatic Bet concentrations [37] in rats. These findings may explain why Bet was highly effective in attenuating the increases in plasma Hcy caused by Met loading in the current study.

Apoptosis plays an essential role in maintaining cellular homeostasis during development, pathophysiological, and differentiation processes in multicellular organisms [38]. HHcy is a pathological condition characterized by an increase in the plasma concentration of total Hcy [39,40]. Numerous clinical and epidemiological studies have indicated that HHcy is an independent risk factor for atherothrombotic disease. In addition, some studies have revealed that Hcy causes endothelial cell dysfunction and induces apoptotic cell death in cell types relevant to atherothrombotic disease, including endothelial cells and smooth muscle cells [41]. In the present study, we found that the percentages of apoptotic cells and cells with necrotic morphology continued to increase as Hcy concentrations increased, although the absolute numbers were reduced in the presence of Bet. Apoptosis has been widely documented to occur in animal and human atherosclerotic lesions, and apoptotic cell death is increased in the atherosclerotic lesions of mice fed HHcy-inducing diets [15]. The data in this study indicate that Met toxicity can lead to apoptotic cell death in cardiomyocytes. From previous work, it is clear that the liver (as well as the kidneys and pancreas) is a target organ of Met toxicity [19]. Kharbanda et al. [42] found that exposure to 7-deaza-adenosine (a potent SAH hydrolase inhibitor) alone induces apoptosis in rat hepatocytes in primary cultures. We have further shown that Bet confers protection against apoptosis, as the absolute numbers of apoptotic cells were reduced in the presence of Bet. This protective effect of Bet against apoptosis has been corroborated in studies from different laboratories in which apoptosis has been induced under various conditions. Barak et al. [43] have shown that Bet decreases intracellular SAH levels by remethylating Hcy, resulting in a significant attenuation of apoptosis. Observations of elevations in intracellular SAH and increases in hepatocyte apoptosis in the context of alcoholic apoptosis have suggested Bet treatment as a modality for the prevention of this condition [44]. In addition, Bet has been effectively used as a treatment agent for patients with inherited genetic disorders related to HHcy, such as cystathionine β-synthase deficiency and methylene tetrahydrofolate reductase deficiency [43]. Although the Bet detoxification group showed no improvement in growth performance in the current study, we were particularly interested in the protection against apoptosis provided by Bet. The hearts of geese in the Bet detoxification group had lower absolute numbers of apoptotic cells than the hearts of geese in the Met toxicity group, showing that Bet administration appears to reverse the biochemical alterations associated with heart apoptosis.

Previous studies have demonstrated that the process of apoptosis is mediated by proteins in the Bcl-2 family and is accomplished through the Fas pathway or the Caspase-dependent apoptotic pathway, which relies on active mitochondrial control [45]. Once the balance between Bcl-2 and Bax is broken, the Caspase-dependent apoptotic pathway can be activated. In this study, Bcl-2 gene expression in goose cardiomyocytes was decreased in the Met toxicity group and the Bet detoxification group, which is consistent with the findings of previous studies. The results revealed that the apoptotic process was excessively activated by Met toxicity and that Bcl-2 family proteins might be involved in this process in cardiomyocytes. However, the expression of Bcl-2 in the Bet detoxification group was not obvious. The possible mechanisms need to be clarified in further studies.

Elucidation of the metabolic events linking excess Met intake to pathology and their dependence on other underlying metabolic processes and pathologic conditions will provide crucial insights into the roles of these important diet-dependent pathways in cardiovascular health and disease. Overall, our results indicated that the supplementation of feed with 1% Met decreased growth performance and increased plasma Hcy levels, indicating that excess Met can lead to HHcy. Bet was able to effectively lower fasting plasma Hcy levels and to prevent plasma Hcy levels from increasing after Met intake. Met toxicity significantly increased the apoptosis rates of cardiomyocytes (*p* < 0.05), but the presence of Bet reduced the apoptosis rates of these cells. The hearts of Bet-supplemented geese had reduced absolute numbers of apoptotic cells, showing that Bet administration appears to reverse the biochemical alterations associated with heart apoptosis.

## Figures and Tables

**Figure 1 animals-10-01642-f001:**
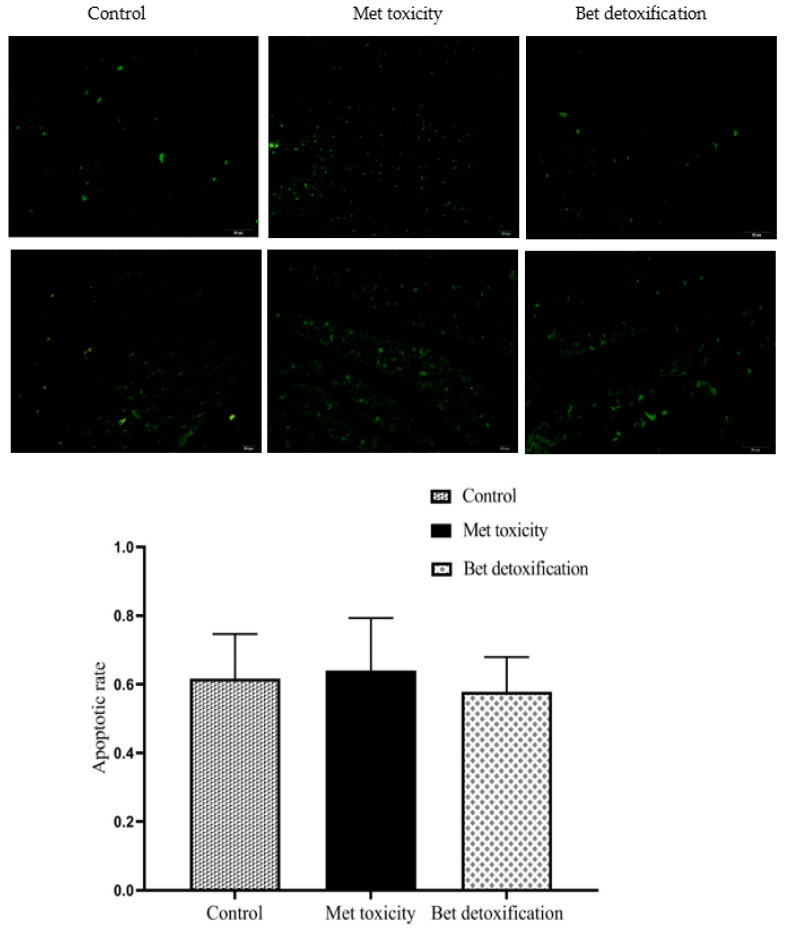
Effect of betaine on cardiomyocyte apoptosis in geese with hyperhomocysteinemia. The control group received the basal diet from 14 to 70 d of age. The methionine (Met) toxicity group received the basal diet supplemented with 1% Met. The betaine (Bet) detoxification group received the Met toxicity diet supplemented with 0.2% Bet.

**Figure 2 animals-10-01642-f002:**
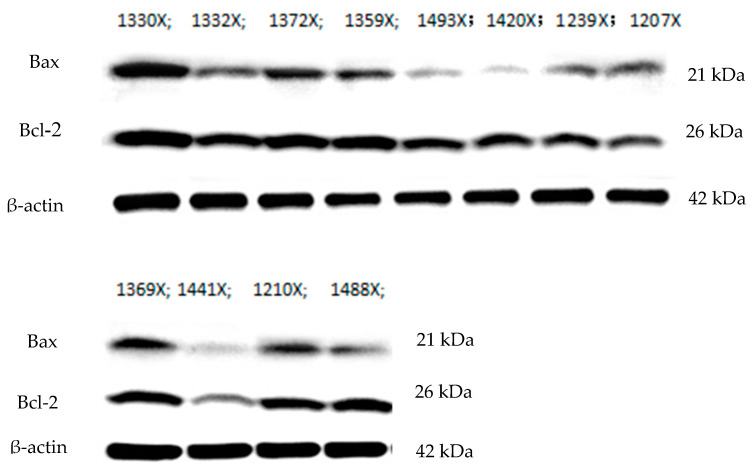
Northern blot analysis of the Bcl-2 associated protein X (Bax) and B-cell lymphoma 2 (Bcl-2) genes in the control group and methionine toxicity group. The numbers 1330X, 1332X, 1372X, 1359X, and 1493X correspond to the control group; the numbers 1493X, 1420X, 1239X, and 1207X correspond to the methionine (Met) toxicity group; and the numbers 1369X, 1441X, 1210X, and 1488X correspond to the betaine (Bet) detoxification group. The control group received the basal diet from 14 to 70 d of age. The Met toxicity group received the basal diet supplemented with 1% Met. The Bet detoxification group received the Met toxicity diet supplemented with 0.2% Bet.

**Table 1 animals-10-01642-t001:** Composition and nutrient levels of the basal diet.

Ingredient	14–28 d	29–70 d
Maize	61.4	64.4
Soybean meal	25.5	23.0
Wheat bran	6.0	2.0
Rice husk	3.5	7.0
Limestone	1.0	1.2
Calcium hydrogen phosphate	1.2	1.0
*DL*-methionine	0.1	0.1
Salt	0.3	0.3
Premix	1.0 ^1^	1.0 ^2^
Total	100.0	100.0
Analyzed nutrient concentrations		
Metabolizable energy ^3^ (MJ/kg)	11.26	11.29
Crude protein (%)	17.00	15.63
Crude fiber (%)	4.30	5.33
Calcium (%)	0.82	0.82
Available phosphorus (%)	0.41	0.37
Methionine (%)	0.35	0.36
Lysine (%)	0.86	0.83
Arginine (%)	1.14	1.13
Histidine (%)	0.44	0.40
Isoleucine (%)	0.73	0.60
Leucine (%)	1.49	1.25
Phenylalanine (%)	0.83	0.74
Threonine (%)	0.68	0.54
Cysteine (%)	0.16	0.25

^1^ The premix was provided by the Yangzhou University Feed Company (Yangzhou, China). One kilogram of premix contained the following: retinol, 1,200,000 IU; cholecalciferol, 400,000 IU; α-tocopherol, 1800 IU; 2-methyl-1,4-naphthoquinone, 150 mg; thiamin, 90 mg; riboflavin, 800 mg; pyridoxine, 320 mg; cobalamin, 1 mg; nicotinic acid, 4.5 g; pantothenic acid, 1100 mg; folic acid, 65 mg; biotin, 5 mg; choline, 45 mg; Fe (as ferrous sulfate), 6 g; Cu (as copper sulfate), 1 g; Mn (as manganese sulfate), 9.5 g; Zn (as zinc sulfate), 9 g; I (as potassium iodide), 50 mg; Se (as sodium selenite), 30 mg. ^2^ One kilogram of premix contained the following: retinol, 1,200,000 IU; cholecalciferol, 400,000 IU; α-tocopherol, 1800 IU; 2-methyl-1,4-naphthoquinone, 150 mg; thiamin, 60 mg; riboflavin, 600 mg; pyridoxine, 200 mg; cobalamin, 1 mg; nicotinic acid, 3 g; pantothenic acid, 900 mg; folic acid, 50 mg; biotin, 4 mg; choline, 35 mg; Fe (as ferrous sulfate), 6 g; Cu (as copper sulfate), 1 g; Mn (as manganese sulfate), 9.5 g; Zn (as zinc sulfate), 9 g; I (as potassium iodide), 50 mg; Se (as sodium selenite), 30 mg. ^3^ The values were calculated from the ingredient apparent metabolizable energy (AME) values for chickens.

**Table 2 animals-10-01642-t002:** Effect of betaine on the growth performance of geese with hyperhomocysteinemia.

Variable		Control	Met Toxicity	Bet Detoxification	*p*-Value
14 d BW (g)		473 ± 0.31	473 ± 0.42	473 ± 0.56	0.826
28 d BW (g)		1459 ± 70 ^a^	1355 ± 40 ^b^	1224 ± 77 ^c^	<0.001
49 d BW (g)		3298 ± 135 ^a^	2910 ± 189 ^b^	2539 ± 107 ^c^	<0.001
70 d BW (g)		4311 ± 131 ^a^	3973 ± 122 ^b^	3820 ± 261 ^b^	0.003
Mortality at 70 d (%)		5.72 ± 7.83	17.14 ± 18.63	8.57 ± 12.78	0.422
14–28 d	ADFI (g)	154 ± 7.02 ^a^	136 ± 16 ^b^	117 ± 9.85^c^	0.001
	ADG (g)	104 ± 4.98 ^a^	97 ± 2.86 ^b^	88 ± 5.50 ^c^	<0.001
	F/G	2.20 ± 0.16	2.16 ± 0.24	2.20 ± 0.16	0.951
29–70 d	ADFI (g)	227 ± 13 ^a^	193 ± 22 ^b^	176 ± 15	0.002
	ADG (g)	68 ± 3.59	62 ± 3.24	62 ± 5.37	0.074
	F/G	3.35 ± 0.32	3.10 ± 0.31	2.87 ± 0.42	0.140

The results are expressed as the means ± SDs. For BW, ADG, ADFI, and F/G, n = 5. ^a,b^ Values in the same row with different lowercase superscripts are significantly different (*p* < 0.05), whereas values with the same or no superscripts are not significantly different (*p* > 0.05). BW, body weight; ADFI, average daily feed intake; ADG, average daily gain; F/G, feed-to-gain ratio. The control group received the basal diet from 14 to 70 d of age. The methionine (Met) toxicity group received the basal diet supplemented with 1% Met. The betaine (Bet) detoxification group received the Met toxicity diet supplemented with 0.2% Bet.

**Table 3 animals-10-01642-t003:** Effect of betaine on serum biochemical indices in geese with hyperhomocysteinemia.

Variable	Control	Met Toxicity	Bet Detoxification	*p*-Value
28 d serum Hcy (µmol/L)	21.58 ± 6.30 ^b^	38.90 ± 11.27 ^a^	33.84 ± 12.61 ^a^	0.003
49 d serum Hcy (µmol/L)	14.82 ± 2.20 ^b^	44.54 ± 5.86 ^a^	42.92 ± 9.54 ^a^	<0.001
70 d serum Hcy (µmol/L)	16.16 ± 2.33 ^b^	30.51 ± 11.08 ^a^	19.75 ± 3.82 ^b^	0.017

The data are presented as the means ± SDs with n = 10 per treatment. ^a,b^ Values in the same row with different lowercase superscripts are significantly different (*p* < 0.05), whereas values with the same or no superscripts are not significantly different (*p* > 0.05). Hcy, homocysteine. The control group received the basal diet from 14 to 70 d of age. The methionine (Met) toxicity group received the basal diet supplemented with 1% Met. The betaine (Bet) detoxification group received the Met toxicity diet supplemented with 0.2% Bet.

**Table 4 animals-10-01642-t004:** Effects of betaine on the protein expression of Bax and Bcl-2 in cardiomyocytes of geese with hyperhomocysteinemia.

Groups	Bax	Bcl-2	Bcl-2/Bax
Control	0.863 ± 0.480	1.128 ± 0.305 ^a^	1.454 ± 0.417
Met toxicity	0.318 ± 0.106	0.696 ± 0.147 ^b^	2.477 ± 1.152
Bet detoxification	0.359 ± 0.217	0.602 ± 0.246 ^b^	1.961 ± 0.670
***p*-Value**	0.062	0.029	0.252

The data are presented as the means ± SDs with n = 10 per treatment. ^a,b^ Values in the same column with different lowercase superscripts are significantly different (*p* < 0.05), whereas values with the same or no superscripts are not significantly different (*p* > 0.05). The control group received the basal diet from 14 to 70 d of age. The methionine (Met) toxicity group received the basal diet supplemented with 1% Met. The betaine (Bet) detoxification group received the Met toxicity diet supplemented with 0.2% Bet.

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
