# Peer review of "Hyperhomocysteinemia Induced by Methionine Excess is Effectively Suppressed by Betaine in Geese"

_animals, 2020, doi:10.3390/ani10091642_

Round 1

Reviewer 1 Report

The manuscript reports the effects of betaine supplementation in alleviating methionine-induced hyperhomocysteinemia in geese. This study is crucial to highlight. The following comments to the authors need to address before the manuscript can be considered for publication.

L37, L40, L46, L53, L56, L58, L215,  : Cite a reference

L39, L40, L42, L43, L50, L53, L58, : What is the animal model?

L40: “the constituent amino acids …” What does it mean?

L40: What is the animal model?

L42: Provide more details

L46: Delete “both”

L47: Which metabolism do you mean?

L49: Define the “acute treatment”

L51: “apoptotic” not “Apoptotic”

L66: provide the full name of these genes and their functions

L67: What is the cutoff for HHcy in gees? Cite a reference

L82-83: Why did you choose these supplementation ratios? Clarify, cite references, provide more details for the source

Table 1: No mention for these two diets in the materials section

L107: How did you calculate these traits?

L115: Describe briefly

L118: Why did you choose heart tissues?

L119: Provide the full name for PBS

L125: Provide the full name for BCA

L127: Provide the full name for PVDF

L128: Why did you choose these antibodies?

L137: Why did you choose t-test to compare between two groups whereas you have 3 groups?

Results section: The authors should prove that the birds in treatment 2 were under HHcy. Otherwise, the manuscript should be rejected if they failed to confirm that with a reference

L218-223: Does not make sense

Author Response

"Please see the attachment.”

Reviewer 2 Report

  1. Introduction: Need more clear rationale for the study, and add a hypothesis.
  2. Method: Define Bax and Bcl-2, and add more description, and how did you add Met and betain? removing corn to make space or add on the top of the basal diets?
  3. Discussion: Need more detailed explanation how Bet reduced Hcy at the later period. This is important.

Round 2

Reviewer 1 Report

The authors addressed all of my comments.